# Studies of Parylene/Silicone-Coated Soft Bio-Implantable Optoelectronic Device

**Gunchul Shin**

School of Materials Science and Engineering, University of Ulsan, 12 Technosaneop-ro 55 beon-gil, Nam-gu, Ulsan 44776, Korea; gunchul@ulsan.ac.kr; Tel.: +82-52-712-8067

**Abstract:** Optogenetics is a new neuroscience technology, consisting of biological technology that activates a nerve by light and engineering technology that transmits light to the nerve. In order to transmit light to the target nerve, fiber optics or light-emitting devices have been inserted into the living body, while the motions or emotions of freely moving animals can be controlled using a wirelessly operated optoelectronic device. However, in order to keep optoelectronic devices small in size and operational for a long time in vivo, the need for a thin but robust protective layer has emerged. In this paper, we developed a protective layer, consisting of Parylene and silicone that can protect soft optoelectronic devices inside saline solution for a long time. A chemical vapor deposited Parylene C film between the polydimethylsiloxane layers showed promising optical, mechanical, and water-barrier properties. We expect that these protective layers can be used as an encapsulation film on bio-implantable devices, including wireless optogenetic applications.

**Keywords:** optogenetics; Parylene; optoelectronics; silicone; polydimethylsiloxane; PDMS

## 1. Introduction

Controlling animal behavior or emotions is one area of neuroscience research. A biological technology that stimulates nerves using external light on behalf of the brain was released in 2005 by Edward Boyden and Karl Deisseroth of Stanford University, and called optogenetics [1]. Materials called opsins, which are proteins that respond to light stimulating nerves by opening an ion channel, were developed and applied to animals to control their behavior through external light stimulation [1,2]. Until now, various opsins and technical tools have been studied to provide light to target nerves in optogenetic research. Subsequent to approaches using optical fiber and laser equipment, a number of wireless optogenetics systems have recently been reported that allow animals to move completely freely [3–11]. They introduced a method using an optoelectronic device, such as a light-emitting diode (LED), that is inserted into a living body instead of the existing optical fiber. The wireless power transmission technology using radio frequency has led to many achievements in the behavioral research on freely moving animals [4–11]. However, in the case of an electronic device inserted into a living body, it has been difficult to test in vivo for a long time due to the infiltration of biological fluid, mechanical damage caused by animal movement, and degradation due to body temperature [4–10]. The materials used for protecting various body-implantable devices, including optoelectronic devices, need to be biofriendly and not interfere with the function of the devices, for example, the optical properties for optogenetic devices. Polyimide (PI) is one of the polymers most used for encapsulation and as a supporting material due to its processability at high temperature (<500 °C) [12]. However, in the case of PI-based protective film, its use is limited by its opacity, the solution processing required, and the need for thermal curing at >200 °C for the fabrication of soft materials-based optoelectronic devices [5,7,12]. Although it is optically superior, epoxy-based material is also limited in use because it is not suitable for low-temperature processes [13]. In addition, epoxy-based materials are accompanied

by various abnormal symptoms such as inflammation due to modulus mismatch with biological tissues (<1 MPa) when inserted in vivo due to the rigid mechanical properties and large modulus (a few GPa) [5,10,13]. To overcome this, silicone-based elastomers, such as polydimethylsiloxane (PDMS), which are biofriendly, optically transparent, and soft materials, have been widely used in bio-insertable devices, but they remain a problem to be solved for the stable, long-term operation of devices against the penetration of biofluid during insertion [5,7,10]. Table 1 summarizes the various properties of biocompatible polymers, polyimide, SU-8, one of the epoxy-based materials, Parylene C, and PDMS. In this paper, we introduced Parylene C, a trademark name of poly(p-xylylene) group polymer, which is biofriendly, optically superior, resistant to liquid penetration, and capable of vapor deposition at room temperature; it was placed between layers of PDMS, one of the silicone elastomers, to fabricate a soft, wireless optoelectronic device. We confirmed that the device with protection layers of PDMS/Parylene C/PDMS can be subjected to an acceleration test for a long time in an experimental environment similar to a living body's environment, including similar temperature, biological fluids, and physical movement. This showed the possibility of it being used as a protective film for various bio-implantable devices that can be inserted into a living body as well as for wireless optogenetic devices.

**Table 1.** Material properties of various biocompatible polymers.

| Properties | Parylene C [13,14] | Polyimide [13–15] | SU-8 [13] | Polydimethylsiloxane [5,7,10,13,14] |
|---|---|---|---|---|
| Elastic modulus | ~3 GPa | 2-9 GPa | ~4 GPa | <1 MPa |
| Mechanical rigidity | Rigid | Rigid | Rigid | Soft, Stretchable |
| Color | Clear | Yellowish, Opaque | Clear | Clear |
| Deposition method | Chemical Vapor Deposition | Solution coating | Solution coating | Solution coating |
| Process Temperature | Room Temperature (R.T.) | ~ 250 °C | ~150 °C | R.T. to 100 °C |
| WVTR [1] | ~0.4 | ~1.4 | ~1 | ~70 |
| Applications | Moisture and chemical barrier | Substrate needs high temperature process | Photoresist | Flexible and stretchable devices |

[1] Water vapor transmission rate (g·mm/m$^2$·day).

## 2. Materials and Methods

### 2.1. Fabrication Procedure

Figure 1 shows the overall device fabrication process. The device is manufactured using slide glass as a temporary holding substrate. To prepare the first PDMS layer, we mixed Sylgard 184 (Dow Corning, Midland, MI, USA) and a curing agent at a 10:1 mass ratio, and set it aside for 1 h at room temperature to remove the bubbles. After coating on slide glass using a spin coater under conditions of 2000 rpm and 30 s, a PDMS thin film with a thickness of about 100 μm formed through heat curing for 1 h on a 110 °C hot plate. After cutting commercial aluminum (Al) foil to a suitable size and placing it on the PDMS, using a laser marking system (Hyosung Laser, Bucheon, Korea), it was etched into a prefabricated design of the antenna and LED connection pattern. After removing unnecessary Al residue with tweezers, we connected a micro-LED chip (C450TR2227, Cree, Durham, NC, USA) using soldering paste (SMD291AX250T5, Chip Quik, Ancaster, ON, USA) and a soldering gun (FX-951, Hakco, Osaka, Japan) with a micro tip (T12-J02, Hakco, Osaka, Japan). The details of the fabrication procedure are presented in Figures A1 and A2 and Video S1. In the same way, a second PDMS thin film was formed using Sylgard 184, and the entire device was removed from the slide glass by infiltration of an ethanol between a PDMS and a glass to reduce the adhesive strength. A chemical vapor deposition (CVD) system (LAVIDA 110, Femtoscience, Hwasung, Korea) was used for coating the Parylene C thin film. Di-chloro-p-xylylene (Daisan Kasei, Chiba, Japan), thermally decomposed at 700 °C, was vapor-deposited onto the device at room temperature. More information about the Parylene CVD system can be found in Figure A3. Finally, the entire device was coated again with a PDMS thin film through a dip-coating method to finish.

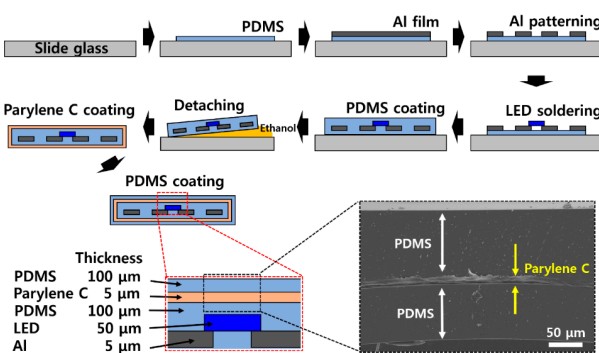

**Figure 1.** Schematic illustration of the fabrication process and side view of layers.

## 2.2. Measurement and Analysis

The thickness of each layer and scanning electron microscope (SEM) images of protection layers are shown at the bottom of Figure 1. Due to the insulating properties of the protective layer, after sputter-coating Au, SEM imaging was performed. The optical characteristics of the manufactured LED device were confirmed using a spectrometer (HR4000, Ocean Optics, Largo, FL, USA) and an integrating sphere. In order to make the experimental environment similar to the in vivo environment, a phosphate buffer solution (1 M, pH 7.4) was prepared and maintained at a temperature of 37, 60, or 90 °C using hot plates and glass bottles with lids, respectively. Near-field communication (NFC)-based wireless power transmission systems, NFC readers (ACR122, Advanced Card System, Kowloon Bay, Hong Kong), and smartphones (Galaxy Note 8, Samsung, Suwon, Korea) were used to operate devices wirelessly, to verify performance, and to verify optical properties.

## 3. Results and Discussion

### 3.1. Fabricated Device

The manufactured optoelectronic device was wirelessly driven through an external NFC reader using a frequency of 13.56 MHz (Figure 2a). A receiving antenna with a diameter of about 18 mm was designed to have a line width of 100 μm, and a line space of about 140 μm was produced by the laser marker. The antenna coil, composed of seven turns, was able to drive the micro-LED by receiving power from the NFC reader without adding an additional impedance matching capacitor. In Figures 2b and A2, the shape of the produced Al antenna pattern was confirmed with a stereo microscope (SDP TOP, Optika, Via Rigla, Italy). The size of the receiving antenna can be changed by adjusting the diameter, line width, line space, and number of turns. When using a double-layered antenna with ultra-thin platform to drive a blue LED with a turn-on voltage of about 3 V in the same wireless system, it has been reported that a receiving antenna of about 8 mm or more is required [8]. In this study, rather than reducing the size of the antenna, a relatively large antenna was used to confirm the protective properties of the coating film.

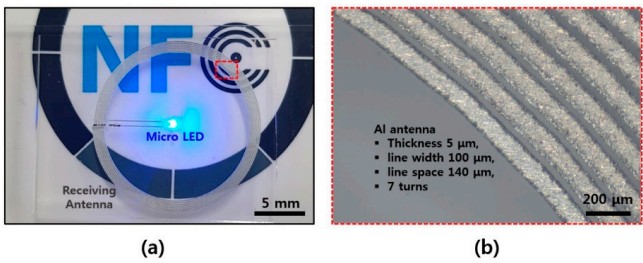

**Figure 2.** Images of fabricated soft optogenetic devices. (**a**) Photograph of wirelessly operated optoelectronic device on near-field communication (NFC) reader; (**b**) microscopic image of laser-cut aluminum (Al) antenna.

### 3.2. Optical Characteristics

The optical properties of optoelectronic devices are shown in Figure 3. In general, the light output power required for optogenetic research is around 10 mW/mm$^2$, but this may vary depending on the type of opsin and the distance between the light source and the target neuron due to the light absorption by biological tissues [1–11]. As shown in Figure 3a, sufficient output power (~50 mW/mm$^2$) was secured even in a low driving environment (3 V, ~5 mA) using the fabricated optoelectronic device here. When a protective film was added onto the LED surface, the light intensity could be decreased due to the optical properties of the film. When Parylene C was deposited on the device, the relative intensity decrease was confirmed by checking the thickness of Parylene C with a spectrometer (Figure 3b). The degree of light transmission varied depending on the type of medium, the roughness of the interface, and the content of impurities in the medium. The blue LED with a wavelength of 450 nm and coated with 5 µm thick Parylene C showed a relative intensity of about 75%, which was similar to results reported before [16]; as the thickness became thicker, the light output power decreased, and it was confirmed that the intensity of the Parylene C film of about 30 µm was 45%. When Parylene C was deposited on the PDMS layer with a thickness of 100 µm, the LED showed a slight decrease in light output that was not significantly different from that of Parylene C only. However, the light intensity increased again when 100 µm of PDMS was coated on the Parylene C/PDMS bilayer. This may be due to the reduced intensity related to light scattering by the surface roughness, which was recovered due to the conformal coating of PDMS on the relatively rough surface of Parylene C. More information about the surface images of Parylene C and PDMS, and energy-dispersive X-ray spectroscopy (EDS) data, can be found in Figure A4.

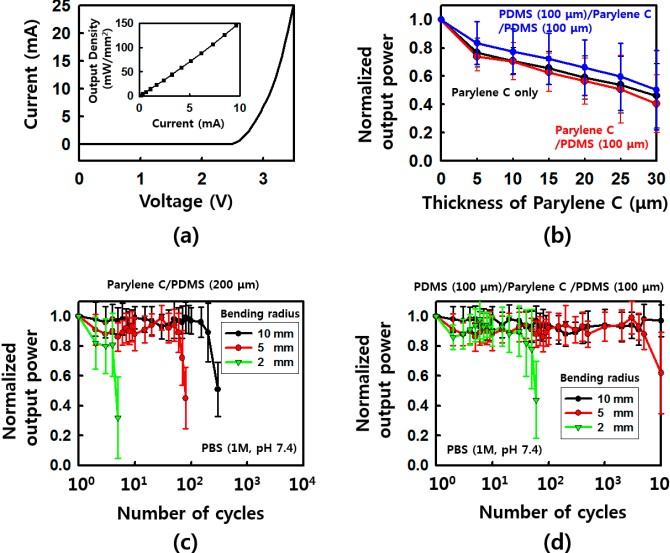

**Figure 3.** Optical properties of protection layer-coated optoelectronic device. (**a**) Graphs of current and voltage, and current and light output; (**b**) normalized light output power as a function of the thickness of Parylene C with/without polydimethylsiloxane (PDMS) layers; (**c**) normalized output power of Parylene C/PDMS-coated devices as a function of the number of bending cycles with different bending radii; (**d**) normalized output power of PDMS/Parylene C/PDMS-coated devices as a function of the number of bending cycles with different bending radii.

### 3.3. Mechanical Stability

When the optogenetic device is inserted into a living body, it is important to secure the maximum operating time since the lifetime of the actual device is significantly reduced due to the physical movements of animals. That is why Parylene C is located in the middle of the PDMS rather than at the top surface. The Parylene C with a relatively large modulus can be mechanically damaged by

tensile stress at the top surface when the device is bent. Therefore, the Parylene C thin film with a large modulus was placed on the mechanical neutral plane [12] located between the thick PDMSs with very low modulus used for flexible or stretchable electronics, so it was not affected by mechanical movements such as bending. The difference between the light output characteristics of the devices that were coated with a bilayer of Parylene C/PDMS or a trilayer of PDMS/Parylene C/PDMS was confirmed using a fatigue test. Figure 3c shows the normalized light output power as a function of bending cycles with a bending radius of 10, 5, or 2 mm in PBS (1 M, pH 7.4). The devices failed to operate after 5, ~80, or ~300 bending cycles when the optoelectronic devices were coated with a Parylene C/PDMS bilayer. The devices that had a trilayer of PDMS/Parylene C/PDMS showed better mechanical stability in PBS during the same fatigue test (Figure 3d). This was due to the tensile strain being focused on the top rigid layer (Parylene C, Y ~3 GPa) and on the top soft layer (PDMS, Y < 1 MPa) during bending deformation, respectively. The soft PDMS layer of the trilayer devices can accommodate the tensile strain and place the Parylene layer in the mechanical neutral plane to prevent the penetration of a saline solution into the electric parts. Details of the bending test are shown in Figure A5.

### 3.4. Water Barrier Properties

In order to confirm the properties of the protective coating of the fabricated device, experimental conditions similar to the in vivo environment were set up. A phosphate buffer solution was prepared to a concentration of 1 M and pH 7.4, and placed in a glass bottle with a lid on a hot plate set at various temperatures. Each bottle was placed on a hot plate to maintain the solution at 37, 60, or 90 °C, and the lid was closed to prevent evaporation of the PBS solution after the device was placed inside the solution. In each bottle, a PDMS (200 µm)-coated device and a PDMS/Parylene C/PDMS (100/5/100 µm)-coated device were put together, and the change in output power intensity was measured every two or five days from the start of the experiment to day 90. Five identical samples were measured for each condition; the average values are given in Figure 4. More information about setting is presented in Figure A6. The device stored in the PBS solution maintained at 37 °C for all samples was stably operated without a significant change in intensity until the 90th day of the last measurement. In the case of the experimental group that set the temperature to 60 and 90 °C, the luminescence properties of the optical element were reduced to about 40% of their initial value after 32 and 7 days, respectively, and the operation of the element was then stopped. Through extrapolation using the acceleration test, how long the device can operate at 37 °C was calculated using the following Arrhenius equation [5,8]:

$$r = A \times \exp(-E_a/kT) \tag{1}$$

where $r$ = the rate of the process, $A$ = a multiplier, $E_a$ = the activation energy for failure, k = Boltzmann constant, and $T$ = process temperature.

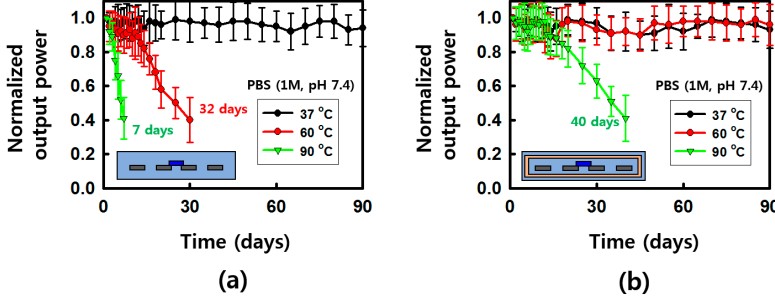

**Figure 4.** Barrier properties of a protection layer-coated optoelectronic device. (**a**) Normalized output power of PDMS-coated devices as a function of immersion time in phosphate buffer solution (PBS) (1 M, pH 7.4) with different temperatures; (**b**) normalized output power of PDMS/Parylene C/PDMS-coated devices as a function of immersion time in phosphate buffer solution (PBS) (1 M, pH 7.4) with different temperatures.

Using the rearranged Arrhenius model, the following equation could be used to calculate the temperature dependence of the time for device failure:

$$\ln(t_2/t_1) = E_a/k \times (1/T_2 - 1/T_1) \tag{2}$$

where $t_2$ = time to failure at $T_2$ and $t_1$ = time to failure at $T_1$.

Calculating using the above equation, it can be predicted that the device can function for up to 120 days, under the assumption that there is no other variable in the case of 37 °C. In the case of a coating film in which 5 μm of a Parylene C thin film was inserted between the PDMS layers, the light emission performance hardly changed after 90 days at 60 °C, and the optical performance was maintained at 40% or more for about 40 days at 90 °C. This shows that the device can be operated at 37 °C for up to ~190 days even if it is assumed that the device at 60 °C stopped working after 90 days, which is known to be a sufficient time for a time test in optogenetic research [5,8]. The time test results with the device that had a thicker layer of Parylene C (10 and 20 μm) between the PDMS layers are shown in Figure A7.

### 3.5. Wireless Operation

The device coated with a protective film composed of PDMS/Parylene C/PDMS exhibits excellent properties both optically and in terms of protection, and is also physically flexible due to its soft material and mechanically neutral structure. Figure 5a shows an image of an optoelectronic device that is wirelessly driven even in bent states (in-folded and out-folded, respectively). The Al antenna of the soft optoelectronic device, even in a bent state, successfully received enough power (>15 mW) to drive a micro-LED on top of the NFC reader. As shown in Figure 5b, a device immersed in PBS at 37 °C for three months still functioned under water because the wireless power transmission uses a radio frequency (RF) of 13.56 MHz, a frequency that is rarely affected by the electrical characteristics of the surrounding environment—unlike cases that use a higher radio frequency [4,5,8]. The optogenetic device is also wirelessly operable with NFC via a commercial smartphone without any special modification (Figure 5c and Video S2). This is expected to be used in personal healthcare applications including optogenetic researches in the near future, such as adding an NFC chip and controlling an animal's behavior or releasing hormones with a personal smartphone platform directly, instead of a large and expensive RF generating system, and without any help from a hospital as well.

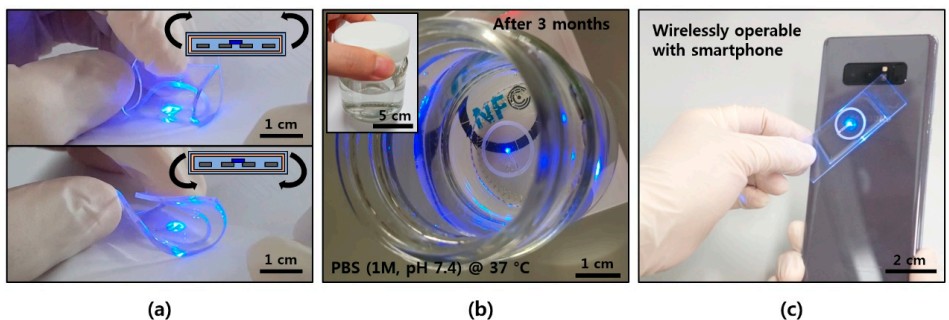

(a)   (b)   (c)

**Figure 5.** Photographs of wirelessly operated optoelectronic devices. (**a**) In-folded (top) and out-folded (bottom) devices with PDMS/Parylene C/PDMS protection layers during wireless operation; (**b**) wireless operation of device after three months immersed in PBS; (**c**) wireless operation of device using NFC of smartphone.

## 4. Conclusions

In this study, a PDMS/Parylene C/PDMS protective film with good optical, physical, and protective properties was designed for a wireless optoelectronic device to be inserted into an animal body for a long time to control the animal's behavior. The device can operate for six months or more in an

environment similar to a living body. We expect that this can be used as an encapsulation layer for various body implantable devices as well as optogenetic studies that control the behavior and emotions of completely free animals.

**Supplementary Materials:** The following are available online at http://www.mdpi.com/2079-6412/10/4/404/s1, Video S1: Al laser cutting, Video S2: wireless operation with smartphone.

**Funding:** This research was partly supported by the National Research Foundation of Korea (grant number NRF-2018R1C1B6006381) and a Korea Evolution Institute of Industrial Technology (KEIT) grant funded by the Korean Government (MOTIE) (No. 10063532, "Development of steel application technologies against ice-induced crashworthiness and artic temperature high toughness").

**Conflicts of Interest:** The author declares no conflict of interest.

## Appendix A

Figure A1 shows the detailed procedure for preparing the optoelectronic device described here. Chip integration and related images are shown in Figure A2. Figure A3 shows the CVD system for the deposition of Parylene. The SEM images and EDS data of Parylene C and PDMS films are presented in Figure A4. The detailed setups for the mechanical test and time test are shown in Figures A5 and A6, respectively. Figure A7 represents the barrier characteristic results with thicker Parylene C between the PDMS layers.

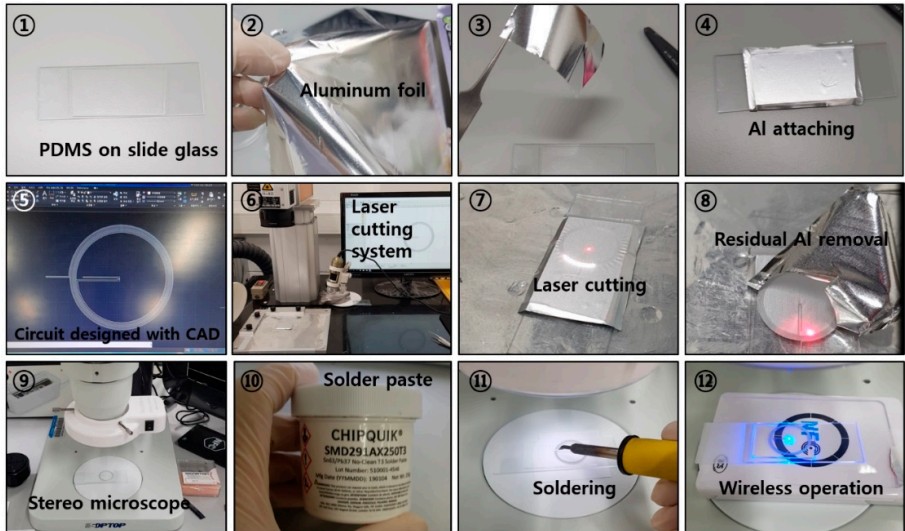

**Figure A1.** Photographs of the fabrication procedure for soft, wirelessly operable optoelectronic device. Aluminum (Al) film was attached (②~④) on top of predeposited PDMS/slide glass substate (①). After etching of Al (⑦) with a laser marker system (⑥) using the design that was prepared with CAD (⑤), residual Al was removed (⑧). LED integration and electrical connection were done with soldering paste (⑩) and a soldering gun (⑪) using a stereo microscope (⑨). The fabricated device was tested wirelessly with an NFC reader (⑫).

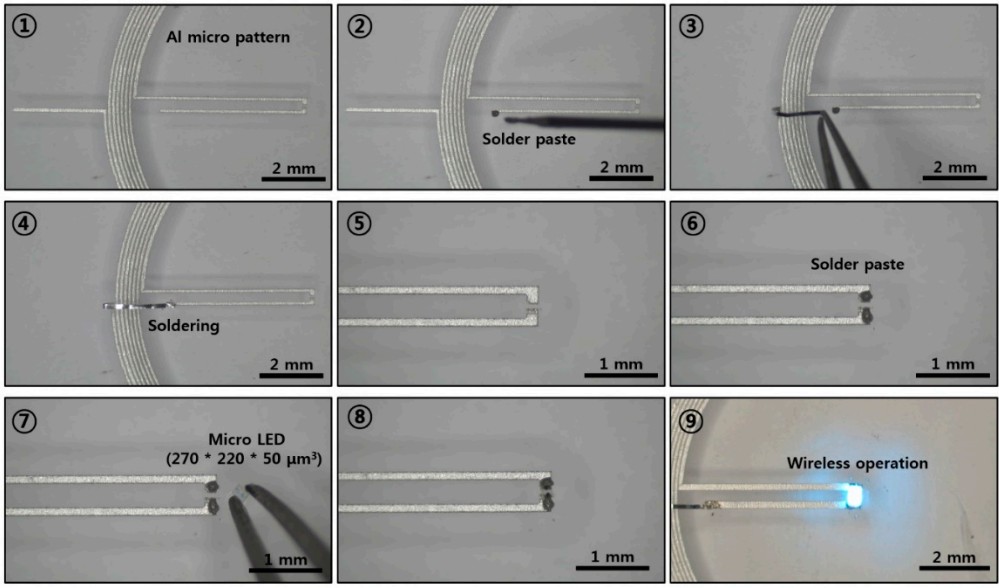

**Figure A2.** Stereomicroscope images of the soldering procedure for chip integration. Solder paste is placed on the parts that will be connected with the antenna and LED chip using a fine needle. The micro-LED was integrated with a soldering gun and micro-tip.

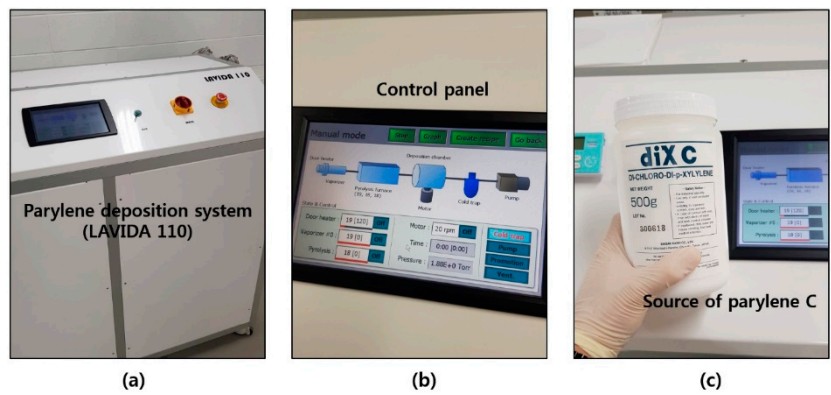

**Figure A3.** Photographs of the chemical vapor deposition (CVD) system for the preparation of Parylene C films: (**a**) Photograph of Parylene CVD (LAVIDA 110); (**b**) photograph of control panel of Parylene CVD; (**c**) photograph of a source material for Parylene C, di-chloro-p-xylylene.

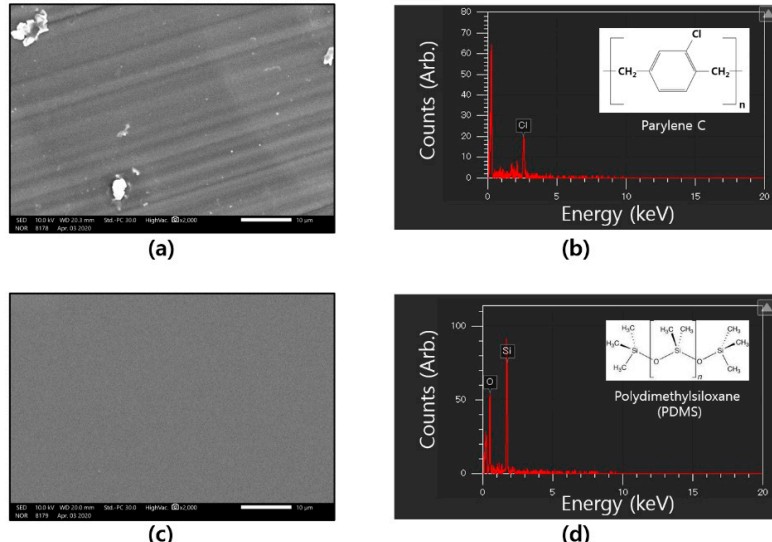

**Figure A4.** Scanning electron microscope (SEM) images and energy-dispersive X-ray spectroscopy data of Parylene C and PDMS films: (**a**) SEM image of chemical vapor deposited Parylene C film; (**b**) EDS data of the surface of Parylene C film; (**c**) SEM image of spin-casted PDMS film; (**d**) EDS data of the surface of PDMS film.

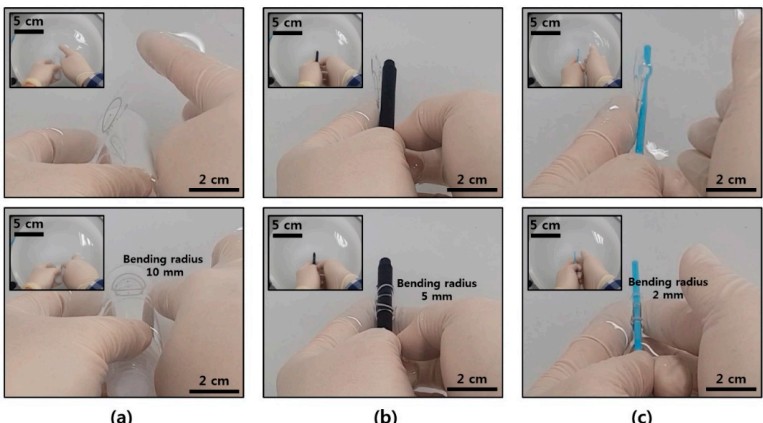

**Figure A5.** Photographs of the setup for mechanical bending tests. All tests proceeded inside saline solution. Images of before (top) and after (bottom) out-folding of devices with a bending radius of (**a**) 10 mm, (**b**) 5 mm, or (**c**) 2 mm.

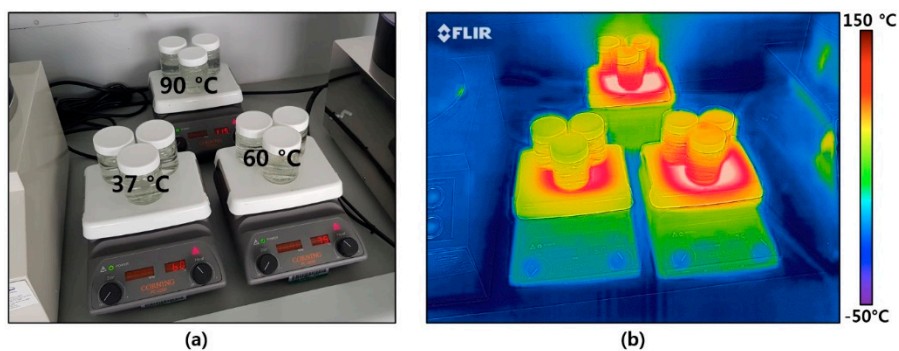

**Figure A6.** Setup for acceleration tests. (**a**) Photograph of the bottles containing various devices on hot plates that were set up to maintain the temperature of bottle at 37, 60, or 90 °C, respectively; (**b**) infrared image of the bottles and hot plates during the time test.

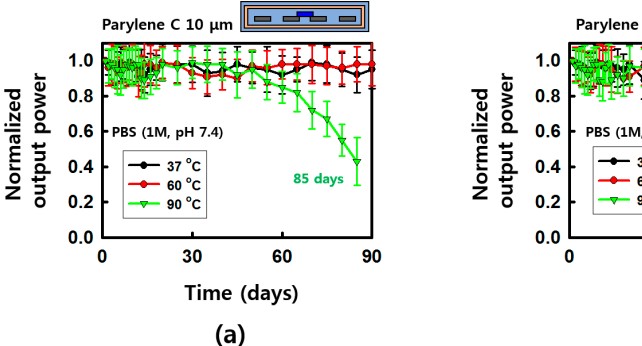

**Figure A7.** Barrier properties of protection layer-coated optoelectronic device. (**a**) Normalized output power of device that has 10 μm of Parylene C between the PDMS layers as a function of immersion time in phosphate buffer solution (PBS) (1 M, pH 7.4) with different temperatures; (**b**) normalized output power of device that has 20 μm of Parylene C between the PDMS layers as a function of immersion time in phosphate buffer solution (PBS) (1 M, pH 7.4) with different temperatures.

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
