# Peer review of "Studies of Parylene/Silicone-Coated Soft Bio-Implantable Optoelectronic Device"

_coatings, doi:10.3390/coatings10040404_

Round 1

Reviewer 1 Report

The manuscript "Studies of Parylene/Silicone-Coated Soft Optoelectronic Implants for Wireless Optogenetic Applications" by Gunchul Shin is a pretty interesting paper that ventures into the relatively new area of animal-device integration. In this work, the device creation and robustness characterization are addressed. This initial work is logical and seems complete, providing a stepping point for future practical studies with the device. The paper is well written, logical and the conclusions are fully supported by the results. Beyond that it does not seem anything is missing, so the work could be accepted in the current form. I can provide a couple of more minor comments on the manuscript. While I think these are minor, maybe the author can use them to improve the work.

1. In section 2.1, how was the glass substrate removed from the soft component of the device? The removal of glass was also not mentioned in the text of 2.1.

2. Figure 5C is a bit weak as it does not show any real physical meaning, so this can probably be omitted.

3. Along with comment 2, what is the distance range that the cellphone can be used to activate the LED?

I hope these comments help the authors with their manuscript.

Reviewer 2 Report

This paper reports a protective layer comprised of parylene and silicone that can protect LED devices inside saline solution. The authors demonstrated that the chemical vapour deposited Parylene C film between two polydimethylsiloxane layers showed promising optical, mechanical and water-barrier properties, and studied its feasibility of wireless operation with a personal smartphone platform. The concept of the work is interesting and should be attractive to most audience, and the characterizations are compressive. In addition, the layout is clear and well shown. Upon addressing the following issue, I would like to support its publication.

The title is somewhat inane. This paper did not perform any measurements regarding the wireless optogenetic applications, albeit that the manufactured parylene-ilicone construction could be used as an encapsulation film on bio-implantable devices in the future. The idea is just an expectation. The authors need to realistically define the title and can mention this promising availability in the outlook section.
